# Exploration and Optimization of the Polymer-Modified NiO_x_ Hole Transport Layer for Fabricating Inverted Perovskite Solar Cells

**DOI:** 10.3390/nano14121054

**Published:** 2024-06-19

**Authors:** You-Wei Wu, Ching-Ying Wang, Sheng-Hsiung Yang

**Affiliations:** Institute of Lighting and Energy Photonics, College of Photonics, National Yang Ming Chiao Tung University, No. 301, Section 2, Gaofa 3rd Road, Guiren District, Tainan 711010, Taiwan; nlpss93048@gmail.com (Y.-W.W.); a.pt11@nycu.edu.tw (C.-Y.W.)

**Keywords:** nickel oxide, perovskite solar cells, polymer additive, poly(vinyl butyral), void-containing

## Abstract

The recombination of charge carriers at the interface between carrier transport layers such as nickel oxide (NiO_x_) and the perovskite absorber has long been a challenge in perovskite solar cells (PSCs). To address this issue, we introduced a polymer additive poly(vinyl butyral) into NiO_x_ and subjected it to high-temperature annealing to form a void-containing structure. The formation of voids is confirmed to increase light transmittance and surface area of NiO_x_, which is beneficial for light absorption and carrier separation within PSCs. Experimental results demonstrate that the incorporation of the polymer additive helped to enhance the hole conductivity and carrier extraction of NiO_x_ with a higher Ni^3+^/Ni^2+^ ratio. This also optimized the energy levels of NiO*_x_* to match with the perovskite to raise the open-circuit voltage to 1.01 V. By incorporating an additional NiO_x_ layer beneath the polymer-modified NiO_x_, the device efficiency was further increased as verified from the dark current measurement of devices.

## 1. Introduction

Perovskite solar cells (PSCs) are widely recognized as one of the most promising photovoltaic technologies in the past decade, owing to their large light absorption coefficients in the visible spectrum, cost-effectiveness, long diffusion length, and facile fabrication [1,2,3,4]. Recently, the advent of organometallic halide PSCs has marked a significant advancement in achieving an impressive photovoltaic conversion efficiency (PCE) of 25.7–26.1% [5,6]. These achievements make PSCs exceptionally valuable for the upcoming generation of solar energy products.

Inverted PSCs, also known as p–i–n structures, are extensively investigated with the utilization of nickel oxide (NiO_x_) as the hole transport layer (HTL) [7,8]. Various techniques, including chemical bath deposition [9], the sol–gel method [10], plasma-assisted atomic layer deposition [11], spray pyrolysis [12], and nanoparticle dispersion [13], have been applied for the production of NiO_x_ HTLs. Given the p-type and hole extraction nature of inorganic NiO_x_, scientists find its widespread use in inverted PSCs, which can be attributed to the existence of Ni vacancies in the lattices accompanying high transmittance in the visible range and environmental stability [3,14]. Although NiO_x_ plays a pivotal role of hole extraction and transport in PSCs, there is still room for hole mobility improvement. As a result, the doping process and/or interfacial modification are employed to enhance hole mobility and extraction capabilities of NiO_x_, thereby reducing carrier recombination and achieving a superior performance of PSCs. To date, the interfacial modification of NiO_x_ films has been implemented through combining NiO_x_ with phthalocyanine or trimercapto-s-triazine trisodium salt [15,16]. On the other hand, transition metal doping such as Cu^2+^ [17,18], Ag^+^ [19], Co^2+^ [20,21], Mn^2+^ [22], and Zn^2+^ [23,24] have proven their effectiveness in enhancing the hole mobility of NiO_x_ films as well as the photovoltaic performance of corresponding PSCs.

From the viewpoint of the mesoscopic junction in PSCs, there have been several studies concerning mesoporous structures of HTLs to improve charge extraction. Wang et al. reported the incorporation of a mesoscopic NiO layer to facilitate hole collection, enabling it to host the perovskite absorber and prevent the degradation of photovoltaic performance [25]. Liu, Shen, and their co-workers successfully utilized electrochemical deposition to form mesoporous NiO_x_ films on FTO glass substrates, reducing carrier recombination and augmenting the photocurrent of devices [26]. Chen et al. deposited mesoporous CuGaO_2_ on the compact NiO_x_ to form a double-layered HTL, as it effectively extracted holes from the perovskite due to the increased contact area at the HTL/perovskite interface [27]. Despite being a promising candidate for hole extraction and transport, surprisingly, there has been limited discussion about the formation of mesoporous NiO_x_ HTLs involving organic polymers for fabricating PSCs.

Herein, we reported the preparation of void-containing NiO_x_ by incorporating poly(vinyl butyral) (PVB) (denoted as p-NiO_x_) as the HTL. The mesoporous p-NiO_x_ layer was obtained through high-temperature calcination at 500 °C, effectively enhancing both the transmittance of NiO_x_ and hole transport within PSCs. To comprehensively investigate the impact of p-NiO_x_ as the HTL in the photovoltaic devices, this study also explored the effects of PVB pretreatment on the interface between NiO_x_ and the perovskite layer. Additionally, the original NiO_x_ film (denoted as o-NiO_x_) and p-NiO_x_/o-NiO_x_ films were prepared for comparative analysis. The experimental results reveal that the valence band (VB) of p-NiO_x_ was shifted downwards compared to o-NiO_x_, which is demonstrated in Section 3.1, resulting in better alignment with the perovskite absorbing layer and a consequent increase in the open-circuit voltage (V_OC_) to 1.01 V. Furthermore, incorporating p-NiO_x_/o-NiO_x_ thin films as the HTL demonstrated superior carrier transport capabilities to ameliorate charge extraction and reduced recombination in photovoltaic devices. While the device based on the o-NiO_x_ HTL exhibited a moderate power conversion efficiency (PCE) of 14.84%, the utilization of the p-NiO_x_/o-NiO_x_ structure resulted in a significantly improved PSC performance with the highest PCE of 16.46%.

## 2. Materials and Methods

Detailed information about the starting materials, preparation of perovskite layers, fabrication of PSCs, and characterization techniques is provided in the Supporting Information. The preparation of the o-NiO_x_ and p-NiO_x_ films is listed as follows. The o-NiO_x_ film was prepared via the sol–gel process. Nickel acetate tetrahydrate (0.124 g), ethanolamine (30 μL), and ethanol (5 mL) were mixed in a sealed glass vial and heated at 70 °C until the solution color became translucent green. For the p-NiO_x_, 30 mg of PVB powder was added to the nickel acetate precursor solution. The two precursor films were deposited individually on the FTO substrates from their solutions via spin coating at 4500 rpm for 30 s under an ambient environment, followed by drying on a hotplate at 80 °C for 10 min. The substrates were then transferred into a tube furnace, heated from room temperature to 500 °C within 90 min in air, and sintered at the final temperature for 1 h to obtain the o-NiO_x_ and p-NiO_x_ films. Furthermore, a p-NiO_x_ layer was deposited on top of the o-NiO_x_ layer to form a p-NiO_x_/o-NiO_x_ structure for comparison.

## 3. Results and Discussion

### 3.1. Characterization of the p-NiO_x_

The surface morphology and thickness of the o-NiO_x_ and p-NiO_x_ films on the FTO substrates were verified via scanning election microscopy (SEM) observation. The o-NiO_x_ film with a thickness of 25 nm is very thin and hence the grains of low-lying FTO are clearly seen, as shown in Figure 1a,c. In Figure 1b,d, the p-NiO_x_ showed uniformly distributed cracks on the surface with a thickness of 25 nm, which is close to that of the o-NiO_x_. The formation of voids is attributed to the thermal degradation of PVB during the calcination process of NiO_x_, which is supposed to increase the light transmittance and surface area of the resulting NiO_x_ layer for the subsequent deposition of perovskite layers. Apart from the SEM observation, atomic force microscopy (AFM) experiments were also carried out to investigate the morphological properties and average roughness (*R*_a_) of o-NiO_x_ and p-NiO_x_ films, as displayed in Figure 1e,f. The FTO grains are clearly observed for both samples; moreover, the o-NiO_x_ has a *R*_a_ value of 14.9 nm, and the p-NiO_x_ possesses a higher *R*_a_ value of 17.7 nm, possibly due to those cavities formed by the removal of PVB in the high-temperature calcination process [28]. Furthermore, X-ray diffraction (XRD) experiments were performed to examine the crystalline phases of the o-NiO_x_ and p-NiO_x_ and the corresponding XRD patterns are revealed in Appendix A in the Supporting Information. Three diffraction peaks of NiO_x_ are located at 2θ = 38.9, 42.5, and 64.5° in both XRD patterns, which corresponds to (111), (200), and (220) planes, respectively [29,30], confirming that the crystalline phase of the NiO_x_ was not altered by PVB pretreatment.

The transmission and absorption spectra of the o-NiO_x_, p-NiO_x_ and p-NiO_x_/o-NiO_x_ films were measured to verify the effect of surface voids on their optical properties, which are depicted in Appendix A. The transmittance of the o-NiO_x_ was observed to be ca. 65% in the range of 350–700 nm. The p-NiO_x_ film has the highest transmittance of 80–90% in the same visible range due to the existence of surface voids, as observed from SEM observation in Figure 1b. High transmittance is beneficial for incident photons to enter devices and to be absorbed by the perovskite absorbing layer. In addition, the p-NiO_x_/o-NiO_x_ possesses a lower transmittance of 70–80% in the same range. This is reasonable since an additional NiO_x_ layer was established below the p-NiO_x_ layer. The absorption spectra of the o-NiO_x_, p-NiO_x_ and p-NiO_x_/o-NiO_x_ films are also displayed in Appendix A, which look similar for the three NiO_x_ films. The Tauc plots of different NiO_x_ films are demonstrated in Appendix A, indicating an optical bandgap of 3.8 eV for the o- NiO_x_ layer and 3.73 eV for the p-NiO_x_ and p-NiO_x_/o-NiO_x_ films, which is close to the previous reports [29,31,32,33].

It is well known that the elemental state of Ni^3+^ (Ni_2_O_3_ species) can provide the nonstoichiometric NiO_x_ with hole transport ability [34,35]. Therefore, the X-ray photoelectron spectroscopy (XPS) measurements were performed to investigate the effect of PVB pretreatment on the Ni^3+^/Ni^2+^ ratio as well as hole transport ability. The Ni 2p_3/2_ XPS spectra of the o-NiO_x_, p-NiO_x_ and p-NiO_x_/o-NiO_x_ films are displayed in Figure 2a−c. According to the previous literature [14,24,36], the multicomponent bands can be well fitted with three different states, including NiO (Ni 2p_3/2_ at 853.8 eV), Ni_2_O_3_ (Ni 2p_3/2_ at 855.3 eV), and a satellite peak of Ni^3+^ (at 856.1 eV). The Ni^3+^/Ni^2+^ ratios for the o-NiO_x_, p-NiO_x_ and p-NiO_x_/o-NiO_x_ films were calculated to be 2.17, 2.78 and 3.45, respectively, showing an apparent increasing Ni^3+^ proportion in the Ni 2p spectra after PVB pretreatment. Thus, the p-NiO_x_ has a better hole-transporting capability than the pristine one [34,37]. Until now, the reason for the increased Ni^3+^/Ni^2+^ ratio up to 3.45 for the p-NiO_x_/o-NiO_x_ remains unclear and more experiments should be implemented, such as electrical measurements of hole-only devices. The O *1s* XPS spectra of the o-NiO_x_, p-NiO_x_ and p-NiO_x_/o-NiO_x_ films are presented in Figure 2d−f, revealing two prominent peaks at around 529 eV (O^2–^ from NiO) and 531 eV (O^2–^ from Ni_2_O_3_) [14,36]. In addition, the O^2–^ peak from NiO shifted from 529.08 (o-NiO_x_) to 528.73 (p-NiO_x_) and 528.63 eV (p-NiO_x_/o-NiO_x_), implying possible interactions between PVB and NiO_x_ via electronic transfer.

To further confirm the effect of PVB pretreatment on the single-carrier mobility and conductivity of NiO_x_, simple devices with three different configurations of FTO/o-NiO_x_/Ag, FTO/p-NiO_x_/Ag, and FTO/p-NiO_x_/o-NiO_x_/Ag devices were fabricated and their current−voltage (I−V) characteristics are illustrated in Figure 3a. The p-NiO_x_ device possesses a larger slope than the o-NiO_x_, meaning that PVB pretreatment can improve the conductivity and charge transport ability of NiO_x_. In addition, the p-NiO_x_/o-NiO_x_ device has the largest slope, indicative of the highest conductivity which is in accordance with XPS results. After calcination, the augmentation of the Ni^3+^ fraction facilitates carrier transport and brings about superior hole conductivity. Subsequently, the hole mobility (*μ_h_*) of these films was approximated from the space charge limited current (SCLC) model defined as follows [38,39,40]:(1)$$J=(9/8)\varepsilon \varepsilon_{o}\mu_{h}(V^{2}/L^{3})$$
where *J* is the current density, *ε_0_* is the vacuum dielectric constant, and *ε* is the relative dielectric constant of NiO_x_ [41]. V is the bias voltage, and *L* is the thickness of the NiO_x_ film (∼25 nm). Figure 3b displays the electrical characteristics derived with the SCLC model of ln(JL^3^/V^2^) versus electric filed (V/L)^0.5^. The p-NiO_x_/o-NiO_x_ structure has the highest *μ_h_* of 1.62 × 10^−2^ cm^2^/Vs, while the *μ_h_* of the o-NiO_x_ and p-NiO_x_ are calculated to be 1.11 × 10^−2^ and 1.22 × 10^−2^ cm^2^/Vs, respectively. The augmented *μ_h_* value of the NiO_x_ HTL is expected to bring on the improvement in PCE and device performance of PSCs [22].

The energy levels and work functions (φ_w_) of the o-NiO_x_, p-NiO_x_ and p-NiO_x_/o-NiO_x_ films were implemented via the ultraviolet photoelectron spectroscopy (UPS) analysis. The UPS spectra of different NiO_x_ films in the high- and low-binding energy regions are shown in Figure 4a. The φ_w_ can be obtained through subtracting the high-binding energy cutoff (around 17 eV) from the photon energy of the He I source (21.22 eV) [42,43]. Therefore, the φ_w_ of the o-NiO_x_, p-NiO_x_ and p-NiO_x_/o-NiO_x_ films is determined to be 4.06, 4.08, and 3.99 eV, respectively. It is known that the work function represents the energy difference between vacuum energy levels and the Fermi level (E_F_) [44,45,46]. The energy difference between the valence band (VB) level and the φ_w_ is associated with the low-binding energy cutoff (around 1 eV) [22]. Ergo, the VB of the o-NiO_x_, p-NiO_x_ and p-NiO_x_/o-NiO_x_ films were calculated to be −5.2, −5.24 and −5.31 eV, respectively. The energy level diagram of the different NiO_x_ and the perovskite layers is depicted in Figure 4b, which is comparable to the previous literature [11,22,24,25]. The alignment of energy levels is crucial for optimizing hole extraction and transport efficiency in PSCs. Reducing the energy barrier between the perovskite layer and HTL would decrease the energy loss during charge transport [27]. The p-NiO_x_/o-NiO_x_ exhibits an obviously downshifted VB level which aligns well with the perovskite layer (VB = −5.4 eV), meaning that better hole extraction can be achieved using PVB pretreatment and consequently a higher V_OC_ is anticipated [18].

### 3.2. Characterization of Perovskite Layers on NiO_x_

To analyze the crystallinity and topography of perovskite layers on different NiO_x_ HTLs, the XRD and SEM experiments were conducted. The corresponding XRD patterns and top-view SEM images of perovskite layers are provided in Appendix A. Several intense diffraction peaks at 2θ = 13.95°, 19.86°, 24.58°, 28.33°, 31.82°, 34.91°, 40.51°, and 43.12° were found, corresponding to the (001), (011), (111), (002), (012), (112), (022), and (003) planes of the perovskite, respectively, which are consistent with the previous literature [47,48,49]. Furthermore, the perovskite grains on the o-NiO_x_, p-NiO_x_ and p-NiO_x_/o-NiO_x_ films appear similar in Appendix A. It is known that the NiO_x_ films remained unaltered after PVB pretreatment (see XRD patterns Appendix A) and PVB was removed during the calcination process, and likewise, there would be no significant change in the morphological structure of the perovskite. To conclude, the XRD patterns and top-view SEM images of perovskites on the three NiO_x_ HTLs look similar, implying that the p-NiO_x_ and p-NiO_x_/o-NiO_x_ structures have little or no effect on the crystalline property and morphology of the perovskite.

Figure 5a displays the steady-state photoluminescence (PL) spectra of the perovskites deposited on the FTO, o-NiO_x_, p-NiO_x_, and p-NiO_x_/o-NiO_x_ films. It can be seen that the perovskite deposited on the FTO substrate has the highest PL emission, while the one on the o-NiO_x_ has a lower PL intensity. According to the previous literature, the decrease in PL intensity means an enhanced charge extraction and transport from the perovskite layer to the HTL [18,22]. It seems odd that the perovskite on the p-NiO_x_ possesses the second strongest PL intensity. It is conjectured that the existence of voids led to direct contact between the perovskite and FTO substrate to reduce the carrier extraction ability of NiO_x_. At the same time, the perovskite on the p-NiO_x_/o-NiO_x_ structure has the lowest PL emission, bringing about the improved photovoltaic performance of PSCs. To further verify the PL results of perovskite films on different NiO_x_ films, the time-resolved PL (TR-PL) decay experiments were carried out and the corresponding PL decay curves are shown in Figure 5b. It is seen that the perovskite coated on the p-NiO_x_/o-NiO_x_ structure possessed the fastest PL decay curve compared with other NiO_x_ films, implying that the hole–electron separation was accomplished more effectively [17]. The TR-PL decay curves were fitted using a biexponential model; the fast decay constant τ*_1_* and slow decay constant τ*_2_* represent the surface recombination and charge recombination in the perovskite bulk, respectively [50,51]. Then, the average carrier lifetime (τ*_avg_*) was estimated from the equation $\tau_{avg}\;$= $\sum \left(A_{i}\tau_{i}^{2}\right)∕\sum \left(A_{i}\tau_{i}\right)$, where A*_i_* and τ*_i_* were deduced from the data of the fitted curve [52,53,54]. All the acquired decay constants τ*_1_*, τ*_2_* and τ*_ayg_* are summarized in Appendix A. The τ*_avg_* was calculated to be 84.13, 45.25, 53.72 and 31.14 ns for the perovskite layers on the FTO, o-NiO_x_, p-NiO_x_, and p-NiO_x_/o-NiO_x_ films, respectively. Since the carrier lifetime is inversely proportional to charge extraction, the p-NiO_x_/o-NiO_x_ structure has the best charge extraction capability among all NiO_x_ films, suggesting the highest device performance of PSCs [11,19].

### 3.3. Device Evaluation

The planar p–i–n PSCs with the architecture of FTO/o-NiO_x_, p-NiO_x_ or p-NiO_x_/o-NiO_x_/perovskite/PC_61_BM+TBABF_4_/PEI/Ag were fabricated and evaluated in this study. The cross-sectional SEM image of the whole device is presented in Appendix A to estimate the thickness of each layer. A thickness of 25, 550, 40, 20 and 100 nm is obtained for the p-NiO_x_, perovskite, TBABF_4_-doped PCBM, and the PEI and Ag electrode, respectively. The thickness of the p-NiO_x_/o-NiO_x_ is approximately double that of the p-NiO_x_ layer. Figure 6a presents the current density−voltage (J−V) curves of PSCs based on the o-NiO_x_, p-NiO_x_ or p-NiO_x_/o-NiO_x_ structures as the HTL under AM 1.5G illumination, and Table 1 summarizes the photovoltaic parameters of all devices including J_SC_, V_OC_, FF, and PCE. The control device using the o-NiO_x_ HTL displayed a moderate PCE of 14.8%, a J_SC_ of 22.7 mA/cm^2^, a V_OC_ of 0.9 V, and an FF value of 72%. The best photovoltaic performance was achieved from the device using the p-NiO_x_/o-NiO_x_ HTL, revealing a PCE of 16.5% which is significantly higher than other devices in this study. The J_SC_, V_OC_ and FF of the device based on the p-NiO_x_/o-NiO_x_ HTL were measured to be 21.5 mA/cm^2^, 1.01 V, and 75%, respectively. As for the device using the p-NiO_x_ HTL, the J_SC_, V_OC_, FF, and PCE are 21.0 mA/cm^2^, 1.01 V, 66%, and 14.2%, respectively. Appendix A depicts the statistical distribution for J_SC_, V_OC_, FF and PCE from 20 individual devices. To realize the hysteresis effect, the J–V curves of devices were measured in the reverse and forward scans and corresponding results are displayed in Appendix A and Appendix A. The hysteresis index (HI) is calculated using the equation HI = (PCE_reverse_ − PCE_forward_)/PCE_reverse_, and the device based on p-NiO_x_/o-NiO_x_ has the smallest HI value of 0.09, indicating that the hysteresis phenomenon is reduced through using the p-NiO_x_/o-NiO_x_ bilayered structure as the HTL. Our PSCs maintained good reproducibility and the device based on the p-NiO_x_/o-NiO_x_ HTL showed relatively higher photovoltaic parameters. The improvement in the device performance can be interpreted from several aspects. As previously discussed in the XPS section, the p-NiO_x_/o-NiO_x_ has the largest Ni^3+^/Ni^2+^ ratio and hole transport ability, leading to the enhanced efficiency of PSCs. In the discussion of UPS experiments, the p-NiO_x_/o-NiO_x_ exhibits the smallest φ_w_ as well as matched energy level with the perovskite absorbing layer, thereby facilitating hole extraction from the perovskite to the HTL. Furthermore, the electrical measurements of the p-NiO_x_/o-NiO_x_ device indicate an elevated *μ_h_* which is beneficial for the carrier transport and PCE of devices. Considering the above aspects, the device using the p-NiO_x_/o-NiO_x_ HTL exhibited the best performance as anticipated. To validate the leakage current of devices, dark current measurements were performed and the corresponding results are displayed in Figure 6b. As mentioned in the previous parts, we assumed that using the p-NiO_x_ HTL would encounter an issue of void formation, which could be verified using dark current measurements. The reverse currents from low to high belong to the devices using p-NiO_x_/NiO_x_, o-NiO_x_, and p-NiO_x_ as the HTL. It is evident that the PSC using p-NiO_x_ has a larger leakage current than that using o-NiO_x_ as the HTL. While the PSC based on p-NiO_x_/NiO_x_ possesses the lowest dark current, it conveys benefits for reducing recombination loss and enhancing carrier transport [48,55]. According to the previous literature [56,57], the values of the series resistance (*R*_s_) and shunt resistance (*R*_sh_) of PSCs can be determined from the voltage dependence of the differential resistance (*R*_diff_) using the equation *R*_diff_ = Δ *V/*Δ *J*, as displayed in Appendix A. The *R*_s_ is determined using the extrapolation of the saturated part of the *R*_diff_−*V* curve toward the interception with the resistance axis. The *R*_sh_ is equal to the differential resistance at a bias of 0 V. It is concluded that the device based on the p-NiO_x_/o-NiO_x_ HTL has the largest *R*_sh_ value of 8.35 kΩcm^2^ among the three PSCs, indicative of the best device performance. Figure 6c shows the integrated current densities and external quantum efficiency (EQE) spectra of devices using o-NiO_x_, p-NiO_x_, and p-NiO_x_/NiO_x_ as the HTL. The results attest that the EQE maximum of the device using p-NiO_x_/NiO_x_ achieved about 79% at 550 nm, being the highest spectral line across the visible range. Furthermore, the integrated current densities of 19.17, 17.7, 19.58 mA/cm^2^ were obtained from the devices based on the o-NiO_x_, p-NiO_x_, and p-NiO_x_/NiO_x_ HTLs, respectively. To explore long-term stability, the unencapsulated PSCs were stored in the nitrogen glovebox and their J–V characteristics under AM 1.5G exposure were measured in ambient air. Figure 6d records the PCE evolution of the PSCs based on the o-NiO_x_, p-NiO_x_, and p-NiO_x_/NiO_x_ HTLs. All devices maintained about 70% of their initial efficiency over a period of 50 days. It is noted that the PCE of the fresh PSC based on the p-NiO_x_/NiO_x_ HTL was 16.4% and it dropped to 13% after 50 days of storage, remaining the best performance among the three devices.

## 4. Conclusions

We have successfully prepared the p-NiO_x_ film with surface voids to increase light transmittance and the interfacial area, facilitating the subsequent deposition of perovskite layers. The p-NiO_x_ HTL exhibited elevated carrier mobility and a downward VB shift, significantly enhancing hole transport behavior and reducing the energy barrier between p-NiO_x_ and the perovskite absorber. On the other hand, the usage of the p-NiO_x_ thin film may encounter direct contact between the perovskite and FTO, as deduced from the result of dark current measurements. Among the three NiO_x_ HTLs, the device based on the p-NiO_x_/o-NiO_x_ HTL possessed the lowest leakage current and the best charge extraction capability. Additionally, the highest V_OC_ of 1.01 V, a PCE of 16.5%, and a good device lifetime of up to 50 days were received, presenting the best performance among the three PSCs.

## Figures and Tables

**Figure 1 nanomaterials-14-01054-f001:**
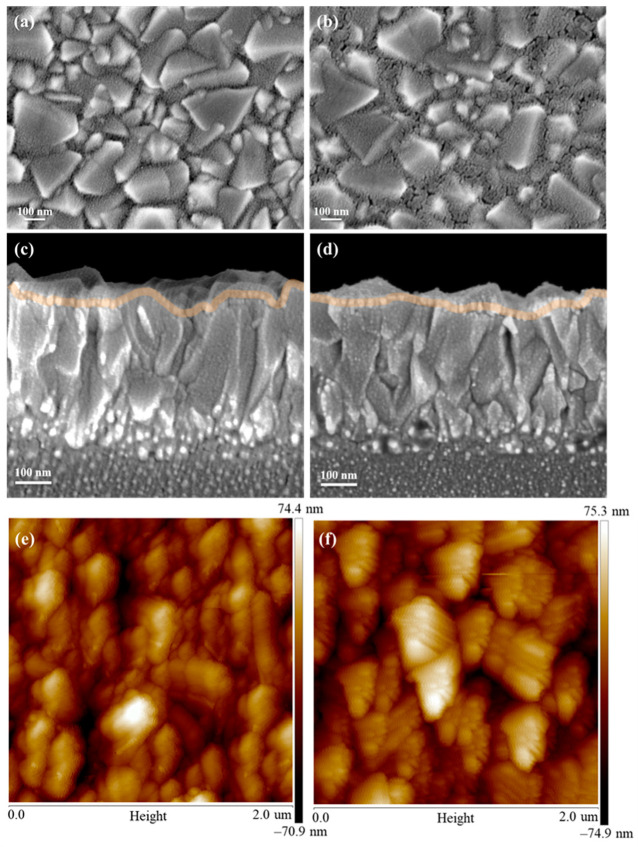
Top-view and cross-sectional SEM images of the (**a**,**c**) o-NiO_x_ and (**b**,**d**) p-NiO_x_ thin films deposited on the FTO substrates; AFM topographic images of the (**e**) o-NiO_x_ and (**f**) p-NiO_x_ thin films.

**Figure 2 nanomaterials-14-01054-f002:**
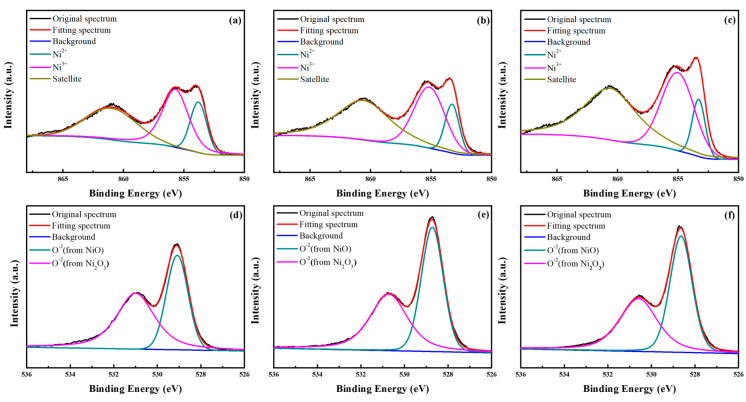
XPS spectra of (**a**–**c**) Ni *2p*_3/2_ and (**d**–**f**) O *1s* elements in the o-NiO_x_, p-NiO_x_ and p-NiO_x_/o-NiO_x_.

**Figure 3 nanomaterials-14-01054-f003:**
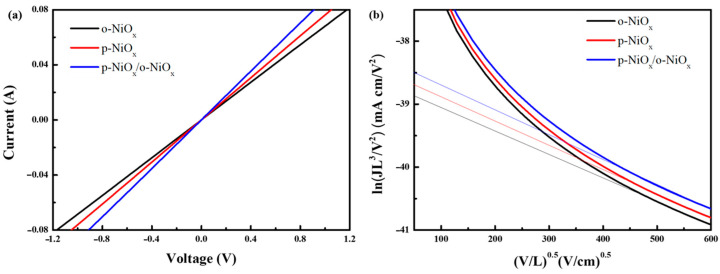
(**a**) Linear sweep voltammetry curves of devices based on the o-NiO_x_, p-NiO_x_ and p-NiO_x_/o-NiO_x_ films; (**b**) hole mobility of the o-NiO_x_, p-NiO_x_ and p-NiO_x_/o-NiO_x_ films versus electric field (V/L)^0.5^.

**Figure 4 nanomaterials-14-01054-f004:**
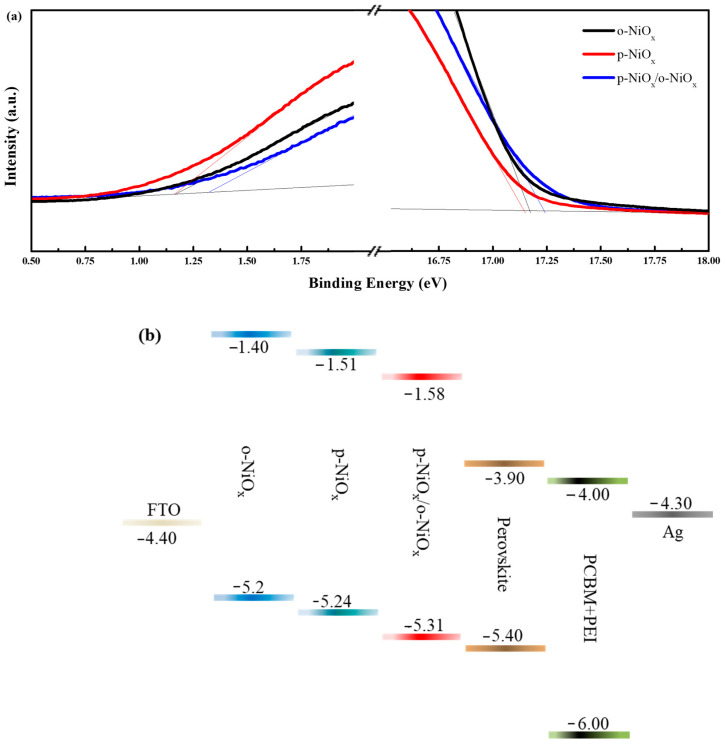
(**a**) UPS spectra of the o-NiO_x_, p-NiO_x_, and p-NiO_x_/o-NiO_x_ films; (**b**) energy level diagram of the whole device (unit: eV).

**Figure 5 nanomaterials-14-01054-f005:**
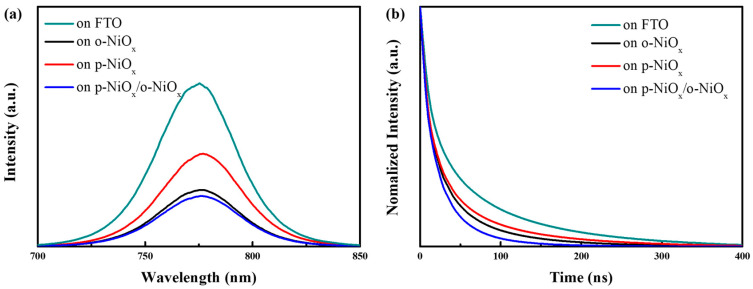
(**a**) PL emission spectra and (**b**) TR-PL decay curves of the perovskites on the FTO, o-NiO_x_, p-NiO_x_ and p-NiO_x_/o-NiO_x_ films.

**Figure 6 nanomaterials-14-01054-f006:**
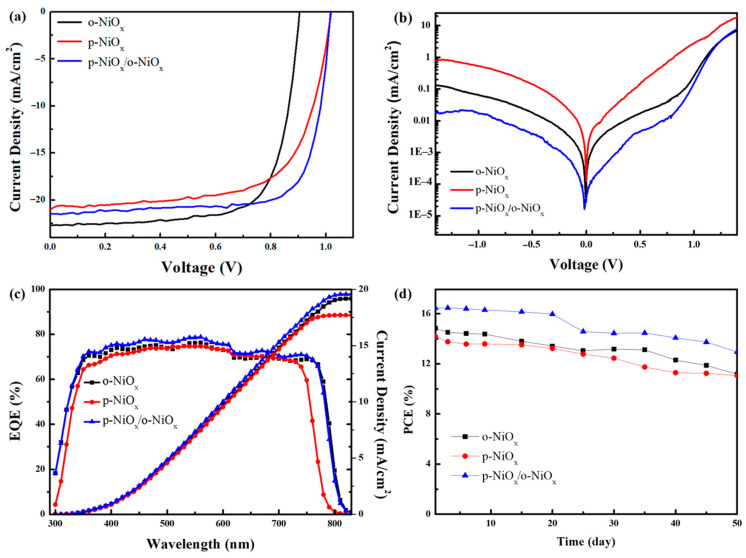
(**a**) J−V characteristics under AM 1.5G exposure, (**b**) dark J−V curves, (**c**) EQE spectra and integrated current density, and (**d**) PCE evolution of devices based on the o-NiO_x_, p-NiO_x_ and p-NiO_x_/o-NiO_x_ HTLs.

**Table 1 nanomaterials-14-01054-t001:** Device performance of inverted PSCs based on the o-NiO_x_, p-NiO_x_ and p-NiO_x_/o-NiO_x_ films as the HTL.

HTL	J_SC_ (mA/cm^2^)	J_SC_ from EQE (mA/cm^2^)	V_OC_ (V)	FF (%)	Best PCE (%)	Avg. PCE ^a^ (%)
o-NiO_x_	22.7	19.2	0.90	72	14.8	14.1
p-NiO_x_	21.0	17.7	1.01	66	14.2	13.5
p-NiO_x_/o-NiO_x_	21.5	19.6	1.01	75	16.5	15.6

^a^ The average PCE was obtained from 20 devices.

## Data Availability

The original contributions presented in the study are included in the article/Appendix A, further inquiries can be directed to the corresponding author.

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
