# Peer review of "Exploration and Optimization of the Polymer-Modified NiOx Hole Transport Layer for Fabricating Inverted Perovskite Solar Cells"

_nanomaterials, 2024, doi:10.3390/nano14121054_

Round 1

Reviewer 1 Report

Comments and Suggestions for Authors

The paper concerns p-i-n perovskite solar cells. The hole transport layer HTL is a modified NiOx layer. The authors showed that by introducing the PVB polymer into NiOx layer and then heating it at a high temperature of 500oC, a porous p-NiOx layer is obtained, which has improved properties due to its use in cells compared to the unmodified layer marked by o-NiOx. p-NiOx layers have greater transmission, greater conductivity and better matched energy levels. The use of a single p-NiOx layer improves Voc, but FF and Jsc decreases significantly, which results in reduced efficiency. Only the use of a p-NiOx/o-NiOx double layer results in an increase in efficiency as a result of improved Voc and FF. The article is interesting for people working with perovskite cells.

However, I have a few questions for the authors.

1. Are there any organic traces left in the p-NiOx layer? Changing morphology alone cannot change the energy levels and mobility of carriers.

2. If some properties of the layer have changed, does the dielectric constant of p-NiOx change and therefore the calculated mobility?

3. Why does the p-NiOx cell with the highest transmission have the lowest current? It should be the biggest. How to explain this fact.

4. Why is the resistance Rs for a cell with a p-NiOx/o-NiOx layer the lowest?

5. Could the authors also show the characteristics for two scanning directions (reverse and forward) and how large the hysteresis is?

6. How explain lower EQE values ​​for long and short waves for p-NiOx?

7. The energy gap determined by the Tauc method for the o-NiOx layer seems have a large error.  The absorption edge is probably greater than 3.73 eV.

In 28: “diffusion distance” should be replaced by  - “diffusion length”.

Author Response

  1. We thank for the reviewer’s comment. We speculated that no organic traces could exist in the p-NiOx layer under 500 oC calcination for 1 hr. According to our research experience, the energy levels and carrier mobility can be altered by changing morphology of materials. For example, in the previous papers DOI:10.3390/nano12193336 and DOI:10.1021/acsomega.1c01378, the nanostructured NiOx exhibited a favorable downward shift of the VB level compared to the NiOx thin film, while the mobility was slightly increased.
  2. Thanks for pointing this out. We agree with the reviewer’s comment that the dielectric constant of NiOx should be changed. However, the variation could be very small. Hence, we put the same dielectric constant value for all NiOx to ensure computational convenience.
  3. The p-NiOx has the highest transmission due to the existence of the void structure, as mentioned in page 4 in the 3.1 Characterization of the p-NiOx section. It is known that device performance of PSCs is determined by many factors, not only the transmittance of materials. About the lowest current of the p-NiOx device, it is conjectured that the existence of voids led to direct contact between the perovskite and FTO substrate to reduce carrier extraction ability of NiOx, as mentioned in the second paragraph in the 3.2 section. Later in the 3.3 section, the dark current measurement of devices was carried out and dark J–V curves are shown in Figure 6(b). It is clearly seen that the PSC using the p-NiOx HTL has the largest leakage current among the three devices, leading to the lowest photogenerated current.
  4. According to another reviewer’s suggestion, we derived the RdiffV curve to obtain Rsh and Rs values more accurately. The Rs can be determined by the extrapolation of the saturated part of the RdiffV curve toward the interception with the resistance axis. The Rsh is equal to the differential resistance at a bias of 0 V. From Figure S8, it can be seen that the device base on the p-NiOx/o-NiOx layer has an Rs of 1.29 Ωcm2, which is not the lowest among the three devices. A new description about the RdiffV curves and R­s values is added in the 3.3 section and two new references [56] and [57] are cited in the revised manuscript. As a result, the old R­s values in Table 1 were removed.
  5. Thanks for the reviewer’s comment. To realize the hysteresis effect, the J–V curves of devices were measured in the reverse and forward scans and corresponding results are displayed in Figure S7 and Table S2. The hysteresis index (HI) is calculated by the equation HI = (PCEreverse – PCEforward)/PCEreverse, and the device based on p-NiOx/o-NiOx has the smallest HI value of 0.09, indicating that the hysteresis phenomenon is reduced by using the p-NiOx/o-NiOx bilayered structure as the HTL. The above description is added in the 3.3 Device evaluation of PSCs section in the revised manuscript.
  6. Thanks for pointing this out. The EQE means incident photon-to-electron conversion efficiency at certain wavelength that is highly dependent on the nature of materials. It could be due to insensitivity of p-NiOx to longer wavelength above 750 nm and shorter wavelength below 350 nm. Frankly, we do not know the exact reason why it has lower EQE values at both sides of visible region.
  7. Thanks for pointing this out. We agree with the reviewer’s comment that the determination of Eg by the Tacu plot for the o-NiOx layer is incorrect. We re-checked the original data and revised the content in Figure S2. The Eg of the o-NiOx is estimated to be 3.80 eV, and therefore its CB is corrected to be -1.40 eV. The description about the optical bandgap in the first paragraph in page 4 is amended. Moreover, the CB level of the o-NiOx layer in Figure 4(b) is also amended.
  8. Thanks for the reviewer’s suggestion. The term “diffusion distance” is replaced by “diffusion length” in the revised manuscript.

Reviewer 2 Report

Comments and Suggestions for Authors

The manuscript „Exploration and optimization of the polymer-modified NiOx hole transport layer for fabricating inverted perovskite solar cells “ by You-Wei Wu, Ching-Ying Wang, and Sheng-Hsiung Yang describes how a modified NiOx hole transporting layer can successfully be employed to improve the performance of inverted perovskite solar cells. The presented results are interesting and could potentially be applied in other areas, where the use of hole transporting layer based on NiOx could be useful, such as other types of perovskite electronics or other types of photovoltaics. There are a few points that the authors should address, before publication in the journal Nanomaterials takes place.

1.       Can the authors clarify, whether any of the NiOx deposition steps described in the Materials and Method Section were performed under inert conditions or whether ambient air is sufficient?

2.       In lines 280 and following, the authors discuss the implications of the results from the dark JV-curves. The authors are encouraged to also perform analysis of the differential resistance R_diff = dV/dJ, which enables them to easily estimate the shunt resistance R_sh and series resistance R_s from the dark JV-curves (see Figure 2 in DOI: 10.1002/aenm.201901438). Thus, it would be more intuitive to assess the potential of the various NiOx-layers over the entire voltage range.

3.       In Table 1, the authors should add the J_sc from EQE integration.

4.       Minor comments related to typos and phrasing can be found in the commented manuscript file.

Author Response

  1. The deposition of NiOx precursor solution was performed under an ambient environment, followed by sintering at 500 oC in air. The above information is added in the 2. Materials and Methods section in the revised manuscript.
  2. We thank for the reviewer’s suggestion. After reading the suggested paper (cited as ref [56]) and another paper concerning differential resistance (cited as ref [57]), we found the two papers useful to the readers and decided to include them in the revised manuscript. According to the previous literature [56,57], the values of series resistance (Rs) and shunt resistance (Rsh) of PSCs can be determined from the voltage dependence of the differential resistance (Rdiff) by the equation Rdiff = Δ V J, as displayed in Figure S8. The Rs is determined by the extrapolation of the saturated part of the RdiffV curve toward the interception with the resistance axis. The Rsh is equal to the differential resistance at a bias of 0 V. It is concluded that the device based on the p-NiOx/o-NiOx HTL has the largest Rsh value of 8.35 kΩcm2 among the three PSCs, indicative of the best device performance. The above description is added in the 3.3 Device evaluation section in the revised manuscript.
  3. Thanks for the reviewer’s suggestion. The Jsc values from EQE integration are added in Table 1 in the revised manuscript.
  4. Thanks for the reviewer’s comment. The typo and phrasing errors have been corrected as follows. Please check the revised manuscript with the “Track Change” function.

    Page 5, line 166, “V is the bias voltage…”

    Page 8, line 269, “…leading to enhanced efficiency of PSCs”

    Page 9, Table 1, the letter “J” is added.

    About the comment “Add scheme with relevant E-levels or mention scheme here in lines 68−71”

    Response: Since the energy level diagram of different NiOx layers is depicted in Figure 4(b) and discussed in the 3.1 section, it would be fine to mention it here. A short description “which is demonstrated in the 3.1 section” is added in the revised manuscript

Round 2

Reviewer 1 Report

Comments and Suggestions for Authors

Thank you for explaining and taking into account my comments.

One editorial error:

Line 190: Figure 4(c b) should be Figure 4(c)

Author Response

Thank you for your approval. About the error that you mentioned, the energy level diagram is depicted in Figure 4(b). There is no Figure 4(c) in the revised manuscript. That's why "Figure 4(c b)" is addressed here.